# Bridging the Gap: Teacher-Assisted Wasserstein Knowledge Distillation for Efficient Multi-Modal Recommendation

## ABSTRACT

Multi-modal recommender systems (MMRecs) leverage diverse modalities to deliver personalized recommendations, yet they often struggle with efficiency due to the large size of modality encoders and the complexity of fusing high-dimensional features. To address the efficiency issue, a promising solution is to compress a cumbersome MMRec into a lightweight ID-based Multi-Layer Perceptron-based Recommender system (MLPRec) through Knowledge Distillation (KD). Despite effectiveness, we argue that this approach overlooks the significant gap between the complex teacher MMRec and the lightweight, ID-based student MLPRec, which differ significantly in size, architecture, and input modalities, leading to ineffective knowledge transfer and suboptimal student performance. To bridge this gap, we propose TARec, a novel teacher-assisted Wasserstein Knowledge Distillation framework for compressing MMRecs into an efficient MLPRec. TARec introduces: (i) a two-staged KD process using an intermediate Teacher Assistant (TA) model to bridge the gap between teacher and student, facilitating smoother knowledge transfer; (ii) logit-level KD using the Wasserstein Distance as metric, replacing the conventional KL divergence to ensure stable gradient flow even with significant teacher-student gaps; and (iii) embedding-level contrastive KD to further distill high-quality embedding-level knowledge from teacher. Extensive experiments on real-world datasets verify the effectiveness of TARec, demonstrating that TARec significantly outperforms the state-of-the-art MMRecs while reducing computational costs. Our anonymous code is available at https://anonymous.4open.science/r/TARec-0980/.

## CCS CONCEPTS

• **Information systems** → **Learning to rank**; *Multimedia information systems.*

## KEYWORDS

Multi-modal Recommendation, Knowledge Distillation, Optimal Transport, Wasserstein Distance

**ACM Reference Format:**
Anonymous Author(s). 2024. Bridging the Gap: Teacher-Assisted Wasserstein Knowledge Distillation for Efficient Multi-Modal Recommendation. In *Proceedings of ACM Conference (Conference'17).* ACM, Singapore, 11 pages. https://doi.org/10.1145/nnnnnnn.nnnnnnn

## 1 INTRODUCTION

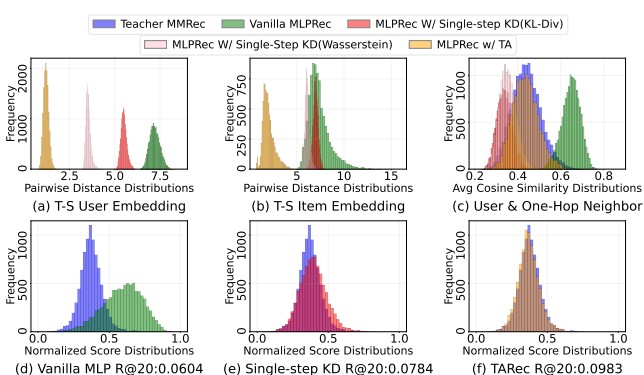

**Figure 1: Visualizations on the distributions of (a)(b): the distances between the teacher and student embeddings for users and items, (c): the average cosine similarity between one-hop neighbourhood nodes, and (d)-(f): the predicted user-item similarity score and the performance with different training methods indicate a significant gap between teacher and student models during knowledge distillation.**

Multi-modal Recommender systems (MMRecs) leverage diverse data from multiple modalities, such as images, text, and audio, to deliver accurate and personalized recommendation [13, 22, 41, 46, 50, 52]. While MMRecs significantly improve recommendation quality, they introduce significant computational overhead due to (1) reliance on large, pre-trained, modality encoders (e.g., CLIP [30] and BERT [6]); (2) the complex architectures required to fuse high-dimensional features from modality encoders [16, 52]. These factors significantly reduce both speed and efficiency, limiting the applicability of MMRecs, especially in industry settings where inference latency and scalability issues are imperative [2, 23].

To address the efficiency issue, a promising approach is to compress a cumbersome MMRec into a lightweight ID-based Multi-Layer Perceptron-based Recommender system (MLPRec) [53] through Knowledge Distillation (KD) [15], thereby optimizing efficiency and hopefully without sacrificing performance. However, we argue that a critical issue with this approach is the large gap between the teacher and student, particularly when they differ significantly in size, architecture and model input. Such a gap often brings distribution shifts between the teacher and the student [29], hinders the student's ability to accurately mimic the teacher, and consequently results in inadequate knowledge transfer and suboptimal model compression [10, 18]. To illustrate this gap, we visualize the pairwise embedding distances between the teacher and student (Figure 1 (a)(b)), the distribution of the similarity between one-hop neighbourhood nodes (Figure 1 (c)) and the distribution of the predicted user-item similarity score (a.k.a., the prediction logits) with

different training methods (Figure 1 (d)-(f), from which we can observe significant gaps at both the embedding and the logit level. Our analysis reveals two key issues associated with this gap:

- **Single-step KD is ineffective for large gap.** While MLPRecs have simple architectures and are less prone to overfitting, they are unable to exploit graph topology and multi-modal features as the MMRecs do. Consequently, the embedding and logit distributions of the MLPRecs can differ significantly from those of MMRecs, as evidenced in Figure 1(a)-(f). Although single-step KD (distilling knowledge directly from the teacher MMRec to the student MLPRec) can reduce this gap to some extent, a notable disparity between teacher and student nevertheless remains, and the student still cannot capture high-quality graph topology from the teacher, as illustrated in Figure 1(a)-(c). This suggests that single-step KD is still insufficient to fully bridge this gap, resulting in misalignment between the student and the teacher and hence suboptimal performance, as evidenced in Figure 1(d)-(f).

- **Limitations of KL Divergence for KD**. While the standard KD adopts the KL Divergence (KL Div.) as metric to measure the divergence between teacher distribution and student distribution, this metric can become problematic when the distribution gap is large, as the KL Div. may produce excessively large values (i.e., not Hölder continuous) and is highly sensitive to even small deformations of the distributions' supports [8], both of which lead to unstable gradients and an increased risk of model collapse [1, 38]. As a result, using KL Div. for KD in our scenario (i.e., a significant gap exists between teacher and student) may not effectively align the distribution of the teacher and with that of the student, leading to inadequate knowledge transfer and degraded performance (Figure 1(a)(b)(d)), while using a Hölder continuous metric (e.g., Wasserstein distance) for KD can significantly reduce the gap both at the embedding and the logit level (Figure 1(a)-(c)).

Building on these insights, we propose TARec, a novel teacher-assistant-enhanced Wasserstein knowledge distillation framework to efficiently compress MMRecs into MLPRec. Specifically, as single-step KD is ineffective to bridge the gap, we introduce an intermediate Teacher Assistant (TA) model, which shares characteristics of both the teacher and the student model, and a two-staged KD process: first distilling the MMRec teacher into the TA, and then distilling the TA into the student MLPRec. In this way, the gaps at both stages (teacher-TA, TA-student) are minimized, facilitating effective knowledge transfer and alignment between the teacher and the student. In light of the limitations of the KL divergence, we propose a novel KD approach based on Wasserstein distance as metric [37]. Unlike KL divergence, Wasserstein distance does not require overlapping distributions to provide stable gradient flow, enabling effective learning even when there is a significant gap between the teacher and student models. Additionally, its continuity regarding convergence in law and ability to metrize topology—i.e., convergence in Wasserstein distance implies weak convergence of distributions—facilitates more precise knowledge transfer at the logit level. [8, 9]. Since students are mainly represented by embedding, we introduce an embedding-level KD loss function to help it absorb collaborative filtering signals from the teacher, effectively distilling high-order signals from user-item interactions into the student's embeddings to enhance performance. Through TARec,

the knowledge from the teacher MMRec is effectively distilled into a simple yet highly efficient MLPRec that is capable of performing fast and accurate inference.

The main contributions of this work are summarized as follows:

- We identify and address the significant gap between multimodal recommendation models and MLP-based student models by proposing the TARec framework, which introduces a teacher assistant model and leverages both logit-level and embedding-level KD to facilitate effective knowledge transfer.
- We propose a novel Wasserstein distance-based KD loss function to overcome the limitations of KL divergence for KD, enabling better distillation of complex collaborative filtering patterns when the gap between the teacher and the student is large.
- We propose an embedding-level contrastive KD loss function to capture collaborative signals in embeddings, allowing the student model to learn user-item interaction relations not fully captured through logit-level distillation.
- Extensive experiments validating the effectiveness of our approach, showing that TARec outperforms the state-of-the-art MMRecs while significantly reducing computational overhead.

## 2 PRELIMINARIES

**Notations and Definitions.** In recommender systems, we denote $\mathcal{U}$ as the set of users and $\mathcal{I}$ as the set of items. The interactions between users and items are represented as a bipartite graph $\mathcal{G} = (\mathcal{V}, \mathcal{E})$, where the node set is $\mathcal{V} = \mathcal{U} \cup \mathcal{I}$ and the edge set $\mathcal{E} \subseteq \mathcal{U} \times \mathcal{I}$ consists of observed user-item interactions. For each user $u \in \mathcal{U}$, we denote $\mathcal{N}_u = \{i \in \mathcal{I} \mid (u, i) \in \mathcal{E}\}$ as the set of items that user $u$ has interacted with. For each item $i \in \mathcal{I}$, we denote $\mathcal{N}_i = \{u \in \mathcal{U} \mid (u, i) \in \mathcal{E}\}$ as the set of users who have interacted with item $i$. Additionally, in multi-modal recommendation, each item $i \in \mathcal{I}$ is associated with a set of features $\mathcal{F}_i = \{f_i^1, f_i^2, \cdots, f_i^M\}$ from $M$ from different modalities, such as texts and images.

**GNN-based Multi-modal Recommendation.** In recommender systems, user-item interactions can be effectively and naturally represented as a bipartite graph. Graph Neural Networks (GNNs) possess the ability to capture complex higher-order relationships by iteratively aggregating information from neighboring nodes [14, 40, 41], establishing GNN-based multi-modal recommendation as a leading approach within Multi-modal Recommender Systems [46, 52]. The typical pipeline for GNN-based Multi-modal Recommender System (MMRec) consists of two main components: modality encoders and neighborhood aggregation layers. Specifically, for each modality $m(1 \leq m \leq M)$, a modality encoder (e.g., CLIP [30] and BERT [6]) extracts the embedding $\mathbf{x}_i^m$ from the modality feature $f_i^m$. For each item $i \in \mathcal{I}$, the set of all its multi-modal embeddings $\{f_i^1, f_i^2, \cdots f_i^M\}$, together with the its ID embedding $\mathbf{e}_i^{ID}$ are fused together [41, 54] to construct its embedding $\mathbf{e}_i \in \mathcal{R}^d$. Additionally, for each user $u \in \mathcal{U}$, its ID embedding is represented as $\mathbf{e}_u \in \mathcal{R}^d$. In the neighborhood aggregation layers, the embeddings of users and items are interatively updated by aggregating information from their neighbors [14]. At each layer $k$, the embeddings are updated as follows:

$$\mathbf{e}_u^{(k+1)} = \sum_{i \in \mathcal{N}_u} \frac{1}{\sqrt{|\mathcal{N}_u|}\sqrt{|\mathcal{N}_i|}} \mathbf{e}_i^{(k)} \quad \mathbf{e}_i^{(k+1)} = \sum_{u \in N_i} \frac{1}{\sqrt{|\mathcal{N}_i|}\sqrt{|\mathcal{N}_u|}} \mathbf{e}_u^{(k)}, \quad (1)$$

where $\mathbf{e}_u^{(k)}$ and $\mathbf{e}_i^{(k)}$ denote the representations of user $u$ and item $i$ at layer $k$, respectively. After $K$ layers of information aggregation, the final representations of users and items are obtained by averaging the representations from all layers, i.e., $\hat{\mathbf{e}}_u = \frac{1}{K+1}\sum_{k=0}^{K}\mathbf{e}_u^{(k)}$, $\hat{\mathbf{e}}_i = \frac{1}{K+1}\sum_{k=0}^{K}\mathbf{e}_i^{(k)}$, and the predicted user-item similarity score (a.k.a., the prediction logits) is computed as $\hat{y}_{ui} = \hat{\mathbf{e}}_u^{\top}\hat{\mathbf{e}}_i$.

**MLP-based Recommendation.** Different from GNN-based MM-Recs that require cumbersome modality feature encoders and graph aggregation layers, MLP-based Recommender system (MLPRec) adopts a simple and lightweight model architecture, and typically do not handle multi-modal features. As a result, they are computationally efficient and less prone to overfitting compared to complex GNNs. Specifically, in MLPRec, the user ID embedding $\mathbf{e}_u \in \mathcal{R}^d$ and the item ID embedding $\mathbf{e}_i \in \mathcal{R}^d$ are simply processed by a multi-layer perceptron to obtain their final representations:

$$\hat{\mathbf{e}}_u = \phi(\mathbf{e}_u) \quad \hat{\mathbf{e}}_i = \phi(\mathbf{e}_i), \quad (2)$$

where $\phi(\cdot)$ represents the multi-layer perceptron. Same as MMRec, the prediction logits for MLPRec is also calculated as $\hat{y}_{ui} = \hat{\mathbf{e}}_u^{\top}\hat{\mathbf{e}}_i$.

**Model Traning.** In the simplest form, both MMRec and MLPRec can be independently trained using the Bayesian Personalized Ranking (BPR) [32] loss function:

$$\mathcal{L}_{\mathrm{bpr}}(\mathcal{H}) = -\sum_{u\in\mathcal{U}}\sum_{i\in\mathcal{N}_u}\sum_{j\notin\mathcal{N}_u}\log\sigma\left(\hat{y}_{u,i} - \hat{y}_{u,j}\right) + \lambda_r\|\Theta_{\mathcal{H}}\|^2, \quad (3)$$

where $\sigma(\cdot)$ denotes the sigmoid function, $\lambda_r$ is the regularization coefficient, and $\Theta_{\mathcal{H}}$ denotes the parameters of model $\mathcal{H}$ (an MMRec or MLPRec) that we are training, $j$ denotes the index of sampled negative items. However, as previously illustrated in Figure 1, independent training leads to significant gap between the teacher MMRec and the student MLPRec, and we will elaborate on how we bridge the gap with our TARec in the following section.

## 3 METHODOLOGY

TARec seamlessly distills knowledge from a cumbersome teacher MMRec into a lightweight student MLPRec via a teacher-assisted, two-staged Wasserstein knowledge distillation approach. Figure 2 provides a framework overview of TARec. We explain the workflow of the Two-staged Teacher-assisted KD process in Section 3.1, and the details of the Wasserstein KD loss function and contrastive KD loss function in Section 3.2 and Section 3.3, respectively. We summarize the training procedure of TARec in Algorithm 1.

### 3.1 Teacher-assisted Two-staged KD

As mentioned earlier, compressing an MMRec into an MLPRec with single-step KD faces a significant gap. The student MLPRec has a very different architecture from the teacher MMRec—it lacks graph topology and does not explicitly utilize multi-modal features—and hence the distribution of prediction logits in the MLPRec may diverge considerably from that of the MMRec. To bridge this gap, we draw inspirations from [26, 34] and introduce a shallower GNN with ID embeddings as input [14] to serve as the Teacher Assistant (TA). The TA $\mathcal{A}$ shares characteristics of both the teacher $\mathcal{T}$ and the student $\mathcal{S}$ by combining the graph topology with ID embeddings, thereby effectively capturing structural information while aligning with the student's focus on ID embeddings.

Our proposed framework, TARec, operates in a two-staged manner. At the first stage, we train a TA network by distilling the teacher into the TA. At the second stage, the TA network then serves as the teacher to guide the student model's training through knowledge distillation (KD). In this way, the gaps at both stages (teacher-TA, TA-student) are minimized, facilitating effective alignment between the teacher and the student. To ensure effective knowledge transfer, we propose a novel logit-level Wasserstein KD loss function $\mathcal{L}_{\mathrm{logit}}$, and an embedding-level contrastive KD loss function $\mathcal{L}_{\mathrm{emb}}$. Details of $\mathcal{L}_{\mathrm{logit}}$ and $\mathcal{L}_{\mathrm{emb}}$ are available in Section 3.2 and Section 3.3 respectively, and we elaborate on the workflow of TARec as follows.

*3.1.1* **Stage One: Teacher-to-Assistant Distillation.** At stage one, we freeze the teacher MMRec $\mathcal{T}$ and distill $\mathcal{T}$ into the TA $\mathcal{A}$ by jointly optimizing the BPR loss function and the KD loss functions:

$$\mathcal{L}_{(\mathcal{T},\mathcal{A})} = \mathcal{L}_{\mathrm{bpr}}(\mathcal{A}) + \lambda_1\mathcal{L}_{\mathrm{logit}}(\mathbf{D}^{\mathcal{T}}, \mathbf{D}^{\mathcal{A}}) + \lambda_2\mathcal{L}_{\mathrm{emb}}(\mathbf{Z}^{\mathcal{T}}, \mathbf{Z}^{\mathcal{A}}), \quad (4)$$

where the logit-level distribution $\mathbf{D}^{\mathcal{T}} = \{\mathbf{y}_{ui}^{\mathcal{T}}|u\in\mathcal{U}, i\in\mathcal{I}\}, \mathbf{D}^{\mathcal{A}} = \{\mathbf{y}_{ui}^{\mathcal{A}}|u\in\mathcal{U}, i\in\mathcal{I}\}$ are constructed using the pairwise ranking score $\mathbf{y}_{ui}^{\mathcal{T}} = \log\sigma((\hat{\mathbf{e}}_u^{\mathcal{T}})^{\top}\hat{\mathbf{e}}_i^{\mathcal{T}} - (\hat{\mathbf{e}}_u^{\mathcal{T}})^{\top}\hat{\mathbf{e}}_j^{\mathcal{T}})$, $\mathbf{y}_{ui}^{\mathcal{A}} = \log\sigma((\hat{\mathbf{e}}_u^{\mathcal{A}})^{\top}\hat{\mathbf{e}}_i^{\mathcal{A}} - (\hat{\mathbf{e}}_u^{\mathcal{A}})^{\top}\hat{\mathbf{e}}_j^{\mathcal{A}})$ in Eq. 3, the embeddings $\mathbf{Z}^{\mathcal{T}} = \{\mathbf{e}_u^{\mathcal{T}}|u\in\mathcal{U}\} \cup \{\mathbf{e}_i^{\mathcal{T}}|i\in\mathcal{I}\}, \mathbf{Z}^{\mathcal{A}} = \{\mathbf{e}_u^{\mathcal{A}}|u\in\mathcal{U}\} \cup \{\mathbf{e}_i^{\mathcal{A}}|i\in\mathcal{I}\}$ are constructed with all the user and item embeddings, the superscript $\mathcal{T}$ and $\mathcal{A}$ denote whether the logits/embeddings come from the teacher or the TA.

*3.1.2* **Stage Two: Assistant-to-Student Distillation.** At stage two, we freeze the TA obtained from stage one, and distill the TA $\mathcal{A}$ into the student MLPRec $\mathcal{S}$ by jointly optimizing the BPR ranking loss function and the KD loss functions:

$$\mathcal{L}_{(\mathcal{A},\mathcal{S})} = \mathcal{L}_{\mathrm{bpr}}(\mathcal{S}) + \lambda_1\mathcal{L}_{\mathrm{logit}}(\mathbf{D}^{\mathcal{A}}, \mathbf{D}^{\mathcal{S}}) + \lambda_2\mathcal{L}_{\mathrm{emb}}(\mathbf{Z}^{\mathcal{A}}, \mathbf{Z}^{\mathcal{S}}), \quad (5)$$

where the logit-level distribution $\mathbf{D}$ and the set of embeddings $\mathbf{Z}$ are constructed in the same way as Eq. 4, the superscript $\mathcal{A}$ and $\mathcal{S}$ denote whether $\mathbf{D}$ and $\mathbf{Z}$ come from the TA or the student.

### 3.2 Logit-level Wasserstein KD

In KD, we need a metric to measure the difference between teacher distribution and student distribution, and some classic metrics include the Kullback-Leibler (KL), reverse KL (RKL), and Jensen-Shannon (JS) divergences, all of which can be viewed as special cases of the *f-divergences* between two distributions $\mathbf{P}$ and $\mathbf{Q}$:

$$\mathcal{D}_f(\mathbf{P}\|\mathbf{Q}) = \int_{x\in\mathcal{X}} f\left(\frac{\mathbf{P}(x)}{\mathbf{Q}(x)}\right)\mathbf{Q}(x)\,dx, \quad (6)$$

where $f$ is a convex function, $\mathcal{X}$ denote the sample space of distributions $\mathbf{P}$ and $\mathbf{Q}$. However, f-divergences are asymmetric and unstable with respect to deformations of the distributions [4, 8]. When two distributions have little or no overlap, these measures encounter training difficulties. For example, the KL divergence may become infinite, and the JS divergence can become locally saturated, leading to vanishing gradients and optimization challenges [1, 4, 7]. Moreover, because f-divergences induce stronger topologies, when the gap between the distribution of the teacher and that of the student is large, they may produce excessively large values (i.e., not Hölder continuous) that lead to unstable gradients and an increased risk of model collapse or mode averaging [1, 38, 47].

**Figure 2: Framework overview of TARec. The left part illustrates the architecture of the MMRec teacher, teacher assistant, and MLPRec student. The middle part details the distillation process, including logit-level Wasserstein KD and embedding-level contrastive KD. The right part outlines the two stages (Teacher-to-Assistant, Assistant-to-student) of the KD process.**

To address this issue, we propose a novel logit-level KD loss function based on Wasserstein distance as metric. Different from f-divergences, Wasserstein distance provides stable gradients even when the distributions have disjoint supports. It induces a weaker topology than f-divergences, enabling smoother optimization and effectively avoiding issues like model collapse associated with f-divergences [1, 8]. We elaborate on the calculation of the Wasserstein KD loss function in Section 3.2.1, and theoretically prove in Section 3.2.2 that the proposed loss function is Hölder continuous, enabling stabilized training even when the distribution gap is large.

*3.2.1* **Wasserstein Distance-Based Knowledge Distillation.**
We denote $(\mathcal{H}, \mathcal{K}) \in \{(\mathcal{T}, \mathcal{A}), (\mathcal{A}, \mathcal{S})\}$ as the model pairs, where we wish to distill model $\mathcal{H}$ into model $\mathcal{K}$. The Wasserstein KD loss function can be formulated as follows:

$$\mathcal{L}_{\text{logit}}(\mathbf{D}^{\mathcal{H}}, \mathbf{D}^{\mathcal{K}}) = W^p\left(\mathbf{D}^{\mathcal{H}}, \mathbf{D}^{\mathcal{K}}\right), \qquad (7)$$

where the Wasserstein distance ($p$-norm) between two probability distributions $\mathbf{P}$ and $\mathbf{Q}$ is defined as follows:

$$W^p(\mathbf{P}, \mathbf{Q}) = \left(\inf_{\pi \in \Gamma(\mathbf{P}, \mathbf{Q})} \int_{X \times Y} d(x, y)^p \, d\pi(x, y)\right)^{1/p}. \qquad (8)$$

Here $\Gamma(\mathbf{P}, \mathbf{Q})$ is the set of all transport plans (couplings) with marginals $\mathbf{P}$ and $\mathbf{Q}$, and $d(x, y)$ denotes the distance between points $x$ and $y$.
**Sinkhorn Algorithm for Calculating Wasserstein Distance.**
Computing the exact Wasserstein distance involves solving a linear programming problem, which is computationally costly and non-differentiable. To address this issue, we employ the Sinkhorn algorithm [5] to transform the problem of calculating Wasserstein distance into calculating an entropy-regularized optimal transport distance $W_\epsilon(\mathbf{P}, \mathbf{Q})$, thereby converting it into a smooth, differentiable convex optimization task. The entropy-regularized optimal transport distance is defined as follows:

$$W_\epsilon(\mathbf{P}, \mathbf{Q}) = \min_{\pi \in \Gamma(\mathbf{P}, \mathbf{Q})} \langle \pi, \mathbf{C} \rangle - \epsilon H(\pi), \qquad (9)$$

where $\pi$ is the transport plan, $\mathbf{C}$ is the cost matrix ($p$-norm) defined as $\mathbf{C}_{ij} = \frac{1}{p}||\mathbf{P}(i) - \mathbf{P}(j)||^p$, $H(\pi) = -\sum_{i,j} \pi_{ij} \log \pi_{ij}$ is the entropy of $\pi$, and $\epsilon$ controls the strength of the regularization. The

Sinkhorn algorithm approximates the optimal transport plan $\pi$ through iterative updates of scaling vectors $\mathbf{u}$ and $\mathbf{v}$:

$$\mathbf{u}^{(\ell+1)} = \frac{\mathbf{P}}{\mathbf{K}\mathbf{v}^{(\ell)}} \quad \mathbf{v}^{(\ell+1)} = \frac{\mathbf{Q}}{\mathbf{K}^\top \mathbf{u}^{(\ell)}}, \qquad (10)$$

where $\mathbf{K}_{ij} = \exp\left(-\frac{\mathbf{C}_{ij}}{\epsilon}\right)$ is the Gibbs kernel matrix computed from the cost matrix $\mathbf{C}$ and the regularization parameter $\epsilon$. The vectors $\mathbf{u}$ and $\mathbf{v}$ are typically initialized as vectors of ones. This iterative process (coordinate ascent) continues until convergence, yielding the optimal transport plan $\pi_{ij} = \mathbf{u}_i \mathbf{K}_{ij} \mathbf{v}_j$. The optimal transport distance is then computed as follows:

$$W_\epsilon(\mathbf{P}, \mathbf{Q}) = \langle \pi, \mathbf{C} \rangle = \sum_{i,j} \pi_{ij} \mathbf{C}_{ij}. \qquad (11)$$

**Unbiased Sinkhorn Divergence.** While the Sinkhorn algorithm enhances computational efficiency through entropy regularization, it introduces an entropic bias. Specifically, for positive $\epsilon$, in general, $W_\epsilon(\mathbf{P}, \mathbf{Q}) \neq 0$, which means that minimizing $W_\epsilon(\mathbf{P}, \mathbf{Q})$ with respect to $\mathbf{P}$ results in a biased solution. To mitigate this, we introduce the unbiased Sinkhorn divergence as the loss function for KD:

$$W_\epsilon^{\text{unbiased}}(\mathbf{P}, \mathbf{Q}) = W_\epsilon(\mathbf{P}, \mathbf{Q}) - \frac{1}{2} W_\epsilon(\mathbf{P}, \mathbf{P}) - \frac{1}{2} W_\epsilon(\mathbf{Q}, \mathbf{Q}). \qquad (12)$$

This formulation subtracts the self-similarity terms, eliminating the entropic bias and ensuring that $\mathcal{W}_\epsilon(\mathbf{P}, \mathbf{Q}) = 0$ when $\mathbf{P} = \mathbf{Q}$ [8]. To demonstrate the issue caused by entropic bias, we visualize in Figure 3 the evolution of the distribution $\mathbf{P}$ toward a fixed distribution $\mathbf{Q}$ [33], guided by the gradient $-\nabla_{\mathbf{x}_i} L(\mathbf{P}, \mathbf{Q})$ (gradient directions shown in red). Both biased and unbiased methods use the same $\epsilon$. The visualization illustrates that, regardless of the gradient flow direction, the speed of loss reduction, or the final convergence results, the unbiased Sinkhorn divergence provides more precise and efficient gradient flows and better alignment, whereas the biased method results in a blurred approximation of the target distribution.

*3.2.2* **Theoretical Analysis.**

THEOREM 1. *The proposed Wasserstein KD loss function is Hölder continuous, i.e., it satisfies the following inequality:*

$$W_p^p(\mathbf{u}, \mathbf{v}) \leq 2^{p-1} L ||\mathbf{u} - \mathbf{v}||_1, \quad p \in [1, \infty]$$

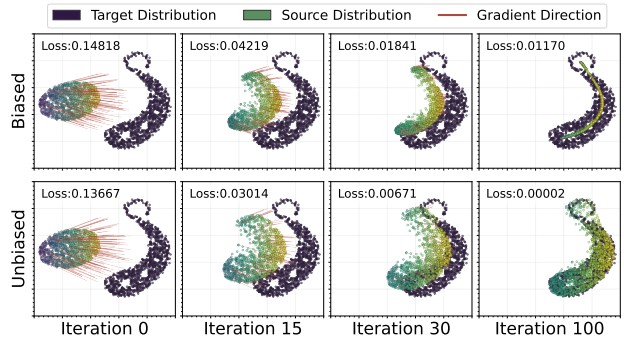

**Figure 3: Evolution of source distribution P towards target Q shows that the unbiased Sinkhorn divergence yields a more efficient and precise gradient flow.**

Detailed proofs are provided in Appendix A.1. This theorem implies that the value of the proposed loss function is upper bounded even when the difference between two distributions **u** and **v** is very large. This property can be very helpful in our setting with a large gap, as it ensures controlled and steady measurement in the difference between the teacher distribution and the student distribution, enabling stable gradient and smooth training process.

## 3.3 Embedding-Level Contrastive KD

Since MLPRec lacks an explicit graph inductive bias for effectively modeling user-item interactions, we introduce an embedding-level contrastive KD loss function to distill high-quality collaborative filtering signals from the teacher model to enhance the corresponding signals in the student's embeddings. Specifically, let $(\mathcal{H}, \mathcal{K}) \in \{(\mathcal{T}, \mathcal{A}), (\mathcal{A}, \mathcal{S})\}$ denote the model pairs, where the goal is to distill $\mathcal{H}$ into $\mathcal{K}$. We employ InfoNCE [27] and construct positive and negative samples from $\mathcal{H}$ and $\mathcal{K}$ based on user-item interaction relations, thereby distilling the high-order collaborative signals into student embeddings. The embedding-level KD loss function is formulated as follows:

$$\mathcal{L}_{\text{emb}}(\mathbf{Z}^{\mathcal{H}}, \mathbf{Z}^{\mathcal{K}}) = -\sum_{u \in \mathcal{U}} \sum_{i \in \mathcal{N}_u} \log \frac{\exp\left((\mathbf{e}_u^{\mathcal{H}})^\top \mathbf{e}_i^{\mathcal{K}}\right)}{(\mathbf{e}_u^{\mathcal{H}})^\top \mathbf{e}_i^{\mathcal{K}} + \sum_{j \notin \mathcal{N}_u} \exp\left((\mathbf{e}_u^{\mathcal{H}})^\top \mathbf{e}_j^{\mathcal{K}}\right)}$$

$$-\sum_{u \in \mathcal{U}} \sum_{i \in \mathcal{N}_u} \log \frac{\exp\left((\mathbf{e}_u^{\mathcal{K}})^\top \mathbf{e}_i^{\mathcal{H}}\right)}{(\mathbf{e}_u^{\mathcal{K}})^\top \mathbf{e}_i^{\mathcal{H}} + \sum_{j \notin \mathcal{N}_u} \exp\left((\mathbf{e}_u^{\mathcal{K}})^\top \mathbf{e}_j^{\mathcal{H}}\right)}.$$

$$(13)$$

## 4 EXPERIMENT

In this section, we conduct comprehensive experiments to answer the following Research Questions (**RQ**s):

- **RQ1:** How does TARec perform compared with the state-of-the-art recommender systems?
- **RQ2:** What is the effectiveness of the key components in TARec?
- **RQ3:** Is TARec efficient in terms of inference latency, memory usage and number of parameters?
- **RQ4:** Can TARec bridge the gap between the teacher MMRec and the student MLPRec?
- **RQ5:** How does TARec perform w.r.t different hyperparameters?

---

**Algorithm 1** The Training Procedure of TARec

---

**Require:** teacher $\mathcal{T}$, teacher assistant $\mathcal{A}$, student $\mathcal{S}$
**Ensure:** a well-trained student MLPRec $\mathcal{S}$
1: **procedure** $W_\epsilon(\mathbf{P}, \mathbf{Q})$
2:     Compute cost matrix $\mathbf{C}$ and kernel matrix $\mathbf{K}$
3:     Initialize vectors $\mathbf{u}$ and $\mathbf{v}$ as vectors of ones
4:     **while** $\mathbf{u}, \mathbf{v}$ do not converge **do**
5:         $\mathbf{u}^{(\ell+1)} = \frac{\mathbf{P}}{\mathbf{K}\mathbf{v}^{(\ell)}} \quad \mathbf{v}^{(\ell+1)} = \frac{\mathbf{Q}}{\mathbf{K}^\top \mathbf{u}^{(\ell)}}$
6:     **end while**
7:     Compute optimal transport plan $\boldsymbol{\pi}_{ij} = \mathbf{u}_i K_{ij} \mathbf{v}_j$
8:     **return** $\sum_{i,j} \boldsymbol{\pi}_{ij} \mathbf{C}_{ij}$
9: **end procedure**
10: Train teacher $\mathcal{T}$ with $\mathcal{L}_{\text{bpr}}(\mathcal{T})$ until convergence
11: **Stage One: Teacher-to-Assistant Distillation**
12: **while** teacher assistant $\mathcal{A}$ do not converge **do**
13:     Compute logits $\mathbf{D}^{\mathcal{T}}, \mathbf{D}^{\mathcal{A}}$
14:     Compute unbiased Sinkhorn divergence:
15:     $\mathcal{L}_{\text{logit}}(\mathbf{D}^{\mathcal{T}}, \mathbf{D}^{\mathcal{A}}) = W_\epsilon(\mathbf{D}^{\mathcal{T}}, \mathbf{D}^{\mathcal{A}}) - \frac{1}{2}W_\epsilon(\mathbf{D}^{\mathcal{T}}, \mathbf{D}^{\mathcal{T}}) - \frac{1}{2}W_\epsilon(\mathbf{D}^{\mathcal{A}}, \mathbf{D}^{\mathcal{A}})$
16:     Compute embedding-level KD loss $\mathcal{L}_{\text{emb}}(\mathbf{Z}^{\mathcal{T}}, \mathbf{Z}^{\mathcal{A}})$
17:     Update teacher assistant $\mathcal{A}$ with $\mathcal{L}_{(\mathcal{T},\mathcal{A})}$
18: **end while**
19: **Stage Two: Assistant-to-Student Distillation**
20: **while** student $\mathcal{S}$ do not converge **do**
21:     Compute logits $\mathbf{D}^{\mathcal{A}}, \mathbf{D}^{\mathcal{S}}$
22:     Compute unbiased Sinkhorn divergence:
23:     $\mathcal{L}_{\text{logit}}(\mathbf{D}^{\mathcal{A}}, \mathbf{D}^{\mathcal{S}}) = W_\epsilon(\mathbf{D}^{\mathcal{A}}, \mathbf{D}^{\mathcal{S}}) - \frac{1}{2}W_\epsilon(\mathbf{D}^{\mathcal{A}}, \mathbf{D}^{\mathcal{A}}) - \frac{1}{2}W_\epsilon(\mathbf{D}^{\mathcal{S}}, \mathbf{D}^{\mathcal{S}})$
24:     Compute embedding-level KD loss $\mathcal{L}_{\text{emb}}(\mathbf{Z}^{\mathcal{A}}, \mathbf{Z}^{\mathcal{S}})$
25:     Update student $\mathcal{S}$ with $\mathcal{L}_{(\mathcal{A},\mathcal{S})}$
26: **end while**

---

## 4.1 Experimental Settings

*4.1.1* **Dataset.** We conduct experiments on three public benchmark multi-modal recommendation datasets. The statistics of datasets are in Appendix A.2, and dataset details are introduced as follows:

- **Netflix:** The Netflix dataset [41] contains user-item interaction records from the Netflix platform. The multimodal content includes movie posters associated with the provided movie titles. Image features are extracted with CLIP-ViT [30], while textual features are encoded using a pre-trained BERT model [19].
- **Tiktok:** The Tiktok micro-video dataset [42] includes user-item interactions and three modality features: visual, acoustic, and textual. The visual and acoustic features are 128-dimensional vectors extracted from micro-videos, and the textual features are obtained from captions using the Sentence-BERT model [31].
- **Electronics:** The Amazon Electronics review dataset [12, 25] contains users' reviews and product information from the electronics domain. The visual modality consists of 4,096-dimensional image features extracted by pre-trained convolutional neural networks [12]. The textual features are generated by combining attributes such as titles, descriptions, categories, and brands into 384-dimensional vectors using the Sentence-BERT model [31].

*4.1.2* **Evaluation Protocols.** In line with previous studies [41, 43], we adopt an all-ranking evaluation strategy to ensure fair comparison [20]. For top-K recommendation tasks, we adopt two widely used metrics, Recall@K (R@K) and Normalized Discounted Cumulative Gain (N@K) with $K = 20$ and 50.

**Table 1: Performance comparisons on benchmark datasets. The best and the second-best performance in each column is bolded and underlined. * indicates the improvements are statistically significant compared to the best baseline (p-value < 0.05).**

| Model | Netflix | | | | Tiktok | | | | Electronics | | | |
|---|---|---|---|---|---|---|---|---|---|---|---|---|
| | R@20 | N@20 | R@50 | N@50 | R@20 | N@20 | R@50 | N@50 | R@20 | N@20 | R@50 | N@50 |
| BPR-MF | 0.1583 | 0.0578 | 0.2396 | 0.0740 | 0.0488 | 0.0177 | 0.1038 | 0.0285 | 0.0211 | 0.0081 | 0.0399 | 0.0117 |
| NGCF | 0.1617 | 0.0612 | 0.2455 | 0.0767 | 0.0604 | 0.0206 | 0.1099 | 0.0296 | 0.0241 | 0.0095 | 0.0417 | 0.0128 |
| LightGCN | 0.1605 | 0.0609 | 0.2449 | 0.0768 | 0.0612 | 0.0211 | 0.1119 | 0.0301 | 0.0259 | 0.0101 | 0.0428 | 0.0132 |
| VBPR | 0.1661 | 0.0621 | 0.2402 | 0.0729 | 0.0525 | 0.0186 | 0.1061 | 0.0289 | 0.0234 | 0.0095 | 0.0409 | 0.0125 |
| MMGCN | 0.1685 | 0.0620 | 0.2486 | 0.0772 | 0.0629 | 0.0208 | 0.1221 | 0.0305 | 0.0273 | 0.0114 | 0.0445 | 0.0138 |
| GRCN | 0.1762 | 0.0661 | 0.2669 | 0.0868 | 0.0642 | 0.0211 | 0.1285 | 0.0311 | 0.0281 | 0.0117 | 0.0518 | 0.0158 |
| LATTICE | 0.1654 | 0.0623 | 0.2531 | 0.0770 | 0.0675 | 0.0232 | 0.1401 | 0.0362 | 0.0340 | 0.0135 | 0.0641 | 0.0184 |
| CLCRec | 0.1801 | 0.0719 | 0.2789 | 0.0892 | 0.0657 | 0.0214 | 0.1329 | 0.0329 | 0.0300 | 0.0118 | 0.0559 | 0.0169 |
| SLMRec | 0.1743 | 0.0682 | 0.2878 | 0.0869 | 0.0669 | 0.0221 | 0.1363 | 0.0342 | 0.0331 | 0.0132 | 0.0624 | 0.0180 |
| BM3 | 0.1792 | 0.0720 | 0.2842 | 0.0923 | 0.0660 | 0.0225 | 0.1351 | 0.0343 | 0.0336 | 0.0141 | 0.0637 | 0.0195 |
| PromptMM | 0.1864 | 0.0743 | 0.3054 | 0.1013 | 0.0737 | 0.0258 | 0.1517 | 0.0410 | 0.0369 | 0.0155 | 0.0691 | 0.0218 |
| TARec | **0.2148*** | **0.0841*** | **0.3189*** | **0.1046*** | **0.0979*** | **0.0369*** | **0.1839*** | **0.0536*** | **0.0490*** | **0.0209*** | **0.0803*** | **0.0271*** |
| Improv. | 15.24% | 13.19% | 4.42% | 3.26% | 32.84% | 43.02% | 21.23% | 30.73% | 32.79% | 34.84% | 16.21% | 24.31% |

*4.1.3* **Implementation Details.** Our TARec is implemented with PyTorch [28]. We employ the AdamW optimizer [24] for training. Learning rates were searched within the range of [1e-4, 1e-3]. The coefficients for $L_2$ weight decay are tuned from {1e-3, 1e-4, 1e-5}. The weight $\lambda_1$ and $\lambda_2$ are tuned from {1e1, 1e0, 1e-1, 1e-2, 1e-3, 1e-4, 1e-5}. The entropy regularization parameter is set as 0.01. All baseline models are evaluated using their respective source codes and original publications, with parameter tuning performed through a unified process to ensure fair comparison.

*4.1.4* **Baselines.** To thoroughly evaluate the performance of TARec, we conduct a comprehensive comparison with several state-of-the-art baselines from two research lines.

i) **Collaborative Filtering Methods**

- **BPR-MF** [32]: It leverages the BPR loss to handle implicit feedback, designed to improve personalized ranking by maximizing the difference between positive and negative interactions.
- **NGCF** [40]: It introduces a GNN-based collaborative filtering framework that injects high-order collaborative filtering signals into representations through embedding propagation layers.
- **LightGCN** [14]: It proposes a simplified graph convolutional network for recommendation by removing redundant designs in the graph convolution layers.

ii) **Multi-Modal Recommendation Methods**

- **VBPR** [13]: It is a matrix factorization-based recommendation approach that integrates visual features from product images to enhance personalized ranking performance.
- **MMGCN** [46]: It captures fine-grained user preferences by constructing modality-specific graphs and refining the representations of users and items for each modality.
- **GRCN** [45]: It introduces an adaptive refinement module that identifies and prunes false positive edges in interaction structures by leveraging multi-modal item characteristics.
- **LATTICE** [51]: It introduces modality-aware structure learning layers and graph convolutions to mine latent collaborative filtering signals from multi-modal content.
- **CLCRec** [44]: It addresses cold-start recommendation by maximizing mutual information between item content and collaborative signals through contrastive learning.

- **SLMRec** [36]: It employs self-supervised learning to capture the multi-modal patterns in recommendation data by generating multiple views of items through data augmentation and applies contrastive learning to improve item representations.
- **BM3** [54]: It introduces a self-supervised multi-modal recommendation framework that bootstraps latent contrastive views from the representations of users and items and jointly optimizes self-supervised multi-modal objective functions.
- **PromptMM** [41]: It enhances multi-modal recommendation through prompt-tuning and single-step KD to compress the MM-Rec into a lightweight MLPRec.

## 4.2 Overall Performance Comparison (RQ1)

Table 1 presents the results of all methods on three datasets. Based on the results, we have the following observations:

- TARec consistently achieves the best performance among all baseline methods. Compared to state-of-the-art multi-modal recommender systems, TARec compresses both modality information and high-order collaborative filtering signals into a simple yet efficient MLPRec, achieving competitive results. Unlike the runner-up method, PromptMM, which uses single-step distillation and employs KL divergence as the metric, we effectively incorporate an intermediate assistant model and a Wasserstein KD loss function, bridging the gap between models. Additionally, embedding-level contrastive KD enhances knowledge transfer during the two-stage training, further boosting performance.
- Compared to traditional matrix factorization methods, graph-based approaches (e.g., NGCF, LightGCN) significantly improve performance. Additionally, MMRecs such as MMGCN and LATTICE, leverage modality content to outperform LightGCN. As indicated in Figure 1 and discussed earlier, vanilla MLPRec is efficient but do not perform well. With TARec, we effectively utilize the advantages of graph-based approaches and the integration of multi-modal content based on an MMRec teacher, achieving effective recommendation results while maintaining efficiency.
- The introduction of multi-modal information generally enhances model performance, but it also presents challenges. For example, the LATTICE method suffers from noise introduced by the homogeneous graph, resulting in poor performance on the Netflix

**Table 2: Ablation study on key components of TARec.**

| Data | Netflix | | Tiktok | | Electronics | |
|---|---|---|---|---|---|---|
| Metrics | R@20 | N@20 | R@20 | N@20 | R@20 | N@20 |
| *w/o*-TA & W. dist. | 0.1796 | 0.0737 | 0.0790 | 0.0296 | 0.0392 | 0.0172 |
| *w/o*-TA | 0.1893 | 0.0749 | 0.0882 | 0.0322 | 0.0419 | 0.0177 |
| *w/o*-W. dist. | 0.2050 | 0.0769 | 0.0888 | 0.0332 | 0.0459 | 0.0201 |
| *w/o*-Emb | 0.2082 | 0.0807 | 0.0954 | 0.0351 | 0.0471 | 0.0203 |
| TARec | **0.2148** | **0.0841** | **0.0979** | **0.0369** | **0.0490** | **0.0209** |

dataset compared to contrastive learning-enhanced methods like CLCRec and SLMRec. Unlike these models, which rely solely on contrastive learning within a single model to refine their embeddings, TARec incorporates embedding-level contrastive KD across different teacher/student model pairs. This approach effectively combines the strengths of both MMRec and contrastive learning through contrastive KD, transferring knowledge from a stronger MMRec model and enhancing the student's embeddings.

### 4.3 Ablation Study (RQ2)

To verify the effectiveness of the key components, we design four variants of TARec:

- *w/o*-TA & W. dist.: This variant directly distills the teacher MM-Rec into the student MLPRec with KL-divergence as metric.
- *w/o*-TA: This variant removes the TA during distillation, while the Wasserstein KD loss remains unchanged.
- *w/o*-W. dist.: This variant replaces the Wasserstein distance with KL-divergence to measure the divergence during distillation.
- *w/o*-Emb: This variant removes the embedding-level contrastive KD loss function during distillation.

The experiment results in Table 2 verify the effectiveness of each component in TARec: (1) For the *w/o*-TA & W. dist. variant, removing both the TA and the Wasserstein distance leads to a significant performance drop across all datasets. This highlights the critical role of these components in bridging the gap between the teacher and student models, indicating that direct knowledge distillation from MMRec using KL-divergence is insufficient. (2) For the *w/o*-TA variant, although the Wasserstein KD is retained to enable better measurement of the differences between the teacher and student models, the student model still struggles to align its multi-modal representations with those of the teacher. This demonstrates the crucial role of the assistant model in narrowing this gap. (3) For the *w/o*-W. dist. variant, although it achieves better results than *w/o*-TA , it still underperforms the full TARec. This suggests that the Wasserstein distance is a more robust and effective metric for capturing and aligning the complex distributional differences between the teacher and student models, especially when there are significant disparities between the two models. (4) For the *w/o*-Emb variant, although it remains competitive with other ablated versions, it still underperforms the full TARec. This demonstrates that the embedding-level contrastive KD plays a crucial role in distilling the higher-order collaborative filtering signal from the teacher into the student. Without it, the model's ability to capture and utilize high-order collaborative filtering semantics across different modalities is weakened, leading to a decline in performance.

**Table 3: Comparisons on model efficiency. "Latency" represents the average inference time for each dataset. "Memory" represents GPU memory usage. "Params." represents the number of model parameters. "Accel." represents the speedup relative to the Teacher model. "Compr." represents the parameter reduction relative to the Teacher model. All experiments are conducted on a single RTX 4090 GPU.**

| Dataset | Model | Latency | Memory | Params. | Accel. | Compr. |
|---|---|---|---|---|---|---|
| Netflix | MMRec | 11.5 ms | 0.93 GB | 29.95M | - | - |
| | MLPRec | 0.33 ms | 0.78 GB | 3.91M | 34.8x | 7.7x |
| Electronics | MMRec | 14.3 ms | 2.32 GB | 104.60M | - | - |
| | MLPRec | 0.34 ms | 1.35 GB | 4.05M | 42.1x | 25.8x |

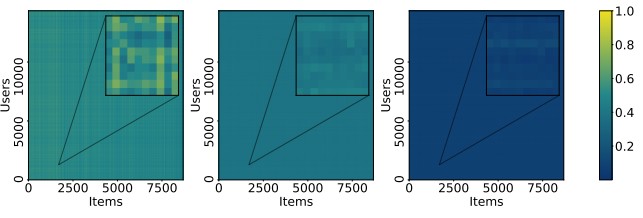

(a) Vanilla MLPRec (b) MLPRec w/ Single-step KD (c) MLPRec w/ TARec

**Figure 4: Heatmaps of the difference of the logits between the teacher and the student from the TikTok dataset.**

### 4.4 Efficiency Analysis (RQ3)

We evaluate the resource utilization and inference efficiency of the teacher MMRec and the student MLPRec in our TARec framework, on the Netflix and Electronics datasets. Table 3 presents the detailed results, focusing on the average inference time, the usage of GPU memory, and the acceleration ratio compared to the teacher model.

The results demonstrate that the compressed student MLPRec is more efficient than the original teacher MMRec in the TARec framework from various perspectives. On the Netflix dataset, TARec reduces the average inference time from 11.5 ms to 0.33 ms, achieving an acceleration ratio of 34.8×. On the Electronics dataset, the time decreases from 14.3 ms to 0.34 ms, resulting in an acceleration ratio of 42.1×. These substantial reductions are attributed to TARec's design: it does not require encoding raw multimodal features from images and text, nor does it rely on modality-based encoders, benefiting from the simple and efficient architecture of the MLPRec. Similarly, TARec demonstrates improved GPU memory efficiency. Memory consumption decreases from 0.93 to 0.78 GB in the Netflix dataset and from 2.32 to 1.35 GB in the Electronics dataset, making TARec suitable for resource-constrained environments.

### 4.5 Qualitative Analysis (RQ4)

To study whether TARec can bridge the gap between the teacher and the student, we visualize the difference of the logit (i.e., $||(\mathbf{e}_u^{\mathcal{T}})^\top \mathbf{e}_i^{\mathcal{T}} - (\mathbf{e}_u^{S})^\top \mathbf{e}_i^{S}||$) between the teacher and the student trained with different methods. From the heatmaps in Figure 4, we observe that the vanilla MLPRec exhibits significant discrepancies in predicting user preferences compared to the teacher MMRec, highlighting the need to bridge this gap. While single-step distillation without TA partially aligns the logit of the two models, MLPRec distilled within the TARec framework achieves the best alignment with the teacher


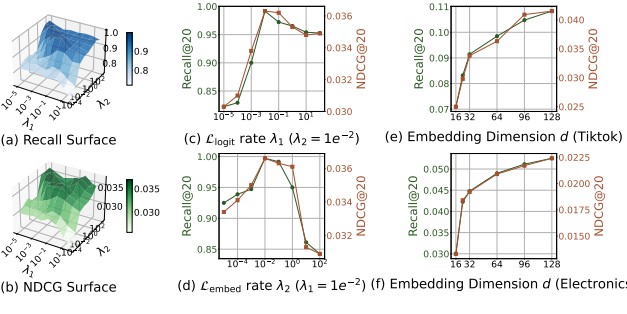

(a) Recall Surface  (c) $\mathcal{L}_{logit}$ rate $\lambda_1$ ($\lambda_2 = 1e^{-2}$)  (e) Embedding Dimension $d$ (Tiktok)

(b) NDCG Surface  (d) $\mathcal{L}_{embed}$ rate $\lambda_2$ ($\lambda_1 = 1e^{-2}$)  (f) Embedding Dimension $d$ (Electronics)

**Figure 5: Impact study of hyperparameters in TARec.**

model's logit, demonstrating the lowest difference with teacher in the heatmap. This indicates that TARec can effectively bridge the gap between the teacher and student, enable the student to learning high-quality collaborative filtering signals from the teacher.

### 4.6 Hyperparameter Sensitivity (RQ5)

We evaluate the influences of key hyperparameters in TARec. From the results in Figure 5, we have the following observations:

- Logit-level KD loss weight $\lambda 1$: We tune $\lambda_1$ from $[10^{-5}, \dots, 10^2]$ and find that both Recall@20 and NDCG@20 increase significantly as $\lambda_1$ rise from $10^{-4}$ to $10^{-2}$, peaking at $\lambda_1 = 10^{-2}$. This indicates that increasing $\mathcal{L}_{logit}$, based on the Wasserstein distance, effectively transfers the teacher model's high-order collaborative filtering signals and modality-related knowledge to the student model. However, beyond $\lambda_1 = 10^{-2}$, performance decreases, suggesting that excessive logit-level alignment can degrade performance in cross-architecture learning.

- Embedding-level contrastive KD loss weight $\lambda 2$: With $\lambda_1$ fixed at $1 \times 10^{-2}$, we tune $\lambda_2$ to assess its impact. Performance of the student model improves as $\lambda_2$ increases up to $1 \times 10^{-2}$ due to effective transfer of the teacher's knowledge at the embedding level. However, unlike $\mathcal{L}_{logit}$, performance sharply declines when $\lambda_2$ exceeds $1 \times 10^{-2}$, because the significant architectural difference between MLPRec and MMRec means that over-aligning the embeddings weakens the performance of MLPRec.

- Joint impact of $\lambda_1$ and $\lambda_2$: We explore the effects of simultaneously varying $\lambda_1$ and $\lambda_2$ across the range $[10^{-5}, \dots, 10^2]$. The results indicate that extremely low values (e.g., $10^{-5}$) for both weights cause a sharp decline in performance, emphasizing the necessity of integrating both logit-level and embedding-level distillation losses. Optimal performance is achieved when $\mathcal{L}_{logit}$ and $\mathcal{L}_{emb}$ are similarly weighted, demonstrating that the model effectively captures complementary knowledge from both levels. Additionally, the model remains robust across a wide spectrum of $\lambda_1$ and $\lambda_2$ values, indicating that TARec is robust to the selection of these hyperparameters and does not require extensive tuning.

- Embedding dimension $d$: We evaluate the impact of varying the embedding dimension $d$ across the range $[16, 32, 64, 96, 128]$. The results show that performance steadily improves as the embedding dimension increases. Performance gains are particularly significant when the dimension ranges from 16 to 64. Beyond 64, although the rate of improvement slows, performance continues to increase. This suggests that, unlike previous MMRec method [41] that may saturate around a dimension of 64, TARec

continues to benefit from higher dimensions, as the TARec framework can leverage stronger teacher models. The positive correlation between embedding dimension and performance enables TARec to support teacher models across a wide range of dimensions, highlighting its strong potential for collaboration with high-dimensional MMRecs based on large modality encoders.

## 5 RELATED WORK

**Multi-Modal Recommender Systems**. Multi-modal recommender systems leverage data from various modalities to provide personalized recommendation, and can alleviate the data sparsity and the cold-start issue. For example, VBPR [13] fuses visual features with item ID embeddings to enhance the performance of collaborative filtering. Subsequently, many works have adopted graph neural network (GNN)-based methods to generate and integrate representations from various modalities, such as MMGCN [46], GRCN [45] and DualGNN [39]. In news recommendation, some studies also integrate multi-modal information like visual content to enhance recommendation performance [48, 49]. Additionally, inspired by advances in contrastive self-supervised learning [3, 11], methods such as CLCRec [44], SLMRec [36] and BM3 [54] employ contrastive learning loss functions to further improve the quality of representation learning. However, these models are inefficienct as they often rely on cumbersome modality encoder and complex fusion layers, which limits their usage in practical applications.

**Knowledge Distillation**. Knowledge distillation aims to transfer knowledge from a complex large model to a smaller model, enabling the latter to achieve comparable performance with fewer computational resources [15]. However, smaller models do not always benefit from stronger teachers [17, 26, 34]. To address this, TAKD [26] introduces an intermediate-sized network to bridge the gap between models, while DGKD [34] collects multiple assistant models to enhance the distillation process. In recommender systems, knowledge distillation is also employed to achieve efficient recommendation, such as DESIGN [35] and PromptMM [41]. However, these works employ a single-step KD approach, which is ineffective to address the large gap arising from the significant difference between the teacher and the student. To bridge this gap, we present a novel teacher-assisted Wasserstein knowledge distillation framework for model compression, enabling effective knowledge transfer and superior performance in multi-modal recommendation.

## 6 CONCLUSION

In this paper, we present TARec, a novel teacher-assisted Wastertein knowledge distillation framework to compress a cumbersome MMRec into an efficient MLPRec. TARec introduces an intermediate-size teacher assistant with a two-staged KD process to bridge the gap between the teacher and the student, develops a novel logit-level Wasserstein knowledge distillation loss function to ensure stable training, and further complement the logit-level KD with an embedding-level contrastive KD to distill the collaborative filtering signals into the embeddings of the student model. Extensive experiments on real-world datasets verify the effectiveness of TARec, demonstrating that TARec significantly outperforms the state-of-the-art MMRecs, reduces computational costs, and effectively bridge the gap between the teacher and the student.

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

# A APPENDIX

## A.1 Proof of Wasserstein Distance Continuity

In TARec, we propose using the Wasserstein distance instead of the KL-divergence to measure the difference in logits between different models, due to its superior continuity properties. Below, we provide a detailed proof of this property.

**Theorem 7.1** Let $\mu$ and $\nu$ be two probability measures on a Polish space $(\mathcal{X}, d)$ with $p \in [1, \infty)$, and fix a point $\mathbf{x}_0 \in \mathcal{X}$. Then:

$$W_p(\mu, \nu) \leq 2^{\frac{1}{p'}} \left( \int_{\mathcal{X}} d(\mathbf{x}_0, \mathbf{x})^p \, d|\mu - \nu|(\mathbf{x}) \right)^{\frac{1}{p}}, \quad \frac{1}{p} + \frac{1}{p'} = 1. \quad (14)$$

**Step 1: Decompose the Measure $\mu - \nu$**

Define the positive and negative parts of the measure $\mu - \nu$:

$$(\mu - \nu)_+ = \max\{\mu - \nu, 0\}, \quad (\mu - \nu)_- = -\min\{\mu - \nu, 0\}. \quad (15)$$

Then, the total variation distance of $\mu - \nu$ is given by:

$$\|\mu - \nu\|_{\text{TV}} = (\mu - \nu)_+ + (\mu - \nu)_-. \quad (16)$$

Define the overlapping part of $\mu$ and $\nu$:

$$\mu \wedge \nu = \mu - (\mu - \nu)_+ = \nu - (\mu - \nu)_-. \quad (17)$$

Define $a$ as the mass of the positive or negative part:

$$a = (\mu - \nu)_+[\mathcal{X}] = (\mu - \nu)_-[\mathcal{X}] \quad (18)$$

**Step 2: Transference Plan $\pi$**

Let $\pi$ be the transference plan obtained by keeping fixed all the mass shared by $\mu$ and $\nu$, and distributing the rest uniformly:

$$\pi = (\text{Id}, \text{Id})_\#(\mu \wedge \nu) + \frac{1}{a} (\mu - \nu)_+ \otimes (\mu - \nu)_-, \quad (19)$$

where $(\text{Id}, \text{Id})_\#(\mu \wedge \nu)$ is the pushforward measure of $\mu \wedge \nu$ under the identity map $(\text{Id}, \text{Id})$, and $\otimes$ denotes the product measure. A more readable version of $\pi$ is:

$$\pi(dx \, dy) = (\mu \wedge \nu)(dx) \, \delta_{y=x}$$
$$+ \frac{1}{a} (\mu - \nu)_+(dx) (\mu - \nu)_-(dy). \quad (20)$$

**Step 3: Estimate the Wasserstein Distance**

According to the definition of the Wasserstein distance, we have:

$$W_p(\mu, \nu)^p \leq \int_{\mathcal{X} \times \mathcal{X}} d(\mathbf{x}, \mathbf{y})^p \, d\pi(\mathbf{x}, \mathbf{y}) = \underbrace{\int_{\mathcal{X}} d(\mathbf{x}, \mathbf{x})^p \, d(\mu \wedge \nu)(\mathbf{x})}_{=0}$$

$$+ \frac{1}{a} \int_{\mathcal{X} \times \mathcal{X}} d(\mathbf{x}, \mathbf{y})^p \, d(\mu - \nu)_+(\mathbf{x}) \, d(\mu - \nu)_-(\mathbf{y}) \quad (21)$$

**Step 4: Apply Inequalities**

Using the triangle inequality $d(\mathbf{x}, \mathbf{y}) \leq d(\mathbf{x}, \mathbf{x}_0) + d(\mathbf{x}_0, \mathbf{y})$ and the convexity inequality $(A + B)^p \leq 2^{p-1}(A^p + B^p)$ for $A, B \geq 0$:

$$d(\mathbf{x}, \mathbf{y})^p \leq [d(\mathbf{x}, \mathbf{x}_0) + d(\mathbf{x}_0, \mathbf{y})]^p$$
$$\leq 2^{p-1} \left( d(\mathbf{x}, \mathbf{x}_0)^p + d(\mathbf{x}_0, \mathbf{y})^p \right). \quad (22)$$

**Step 5: Estimate the Upper Bound of the Integral**

Substituting the inequality into the integral, we get:

$$W_p(\boldsymbol{\mu}, \boldsymbol{\nu})^p \leq \int_{X \times X} d(\mathbf{x}, \mathbf{y})^p \, d\pi(\mathbf{x}, \mathbf{y})$$

$$= \frac{1}{a} \int_{X \times X} d(\mathbf{x}, \mathbf{y})^p \, d(\boldsymbol{\mu} - \boldsymbol{\nu})_+ (\mathbf{x}) \, d(\boldsymbol{\mu} - \boldsymbol{\nu})_- (\mathbf{y})$$

$$\leq \frac{2^{p-1}}{a} \int_{X \times X} \underbrace{\left[ d(\mathbf{x}, \mathbf{x}_0)^p + d(\mathbf{x}_0, \mathbf{y})^p \right] \, d\lambda(\mathbf{x}, \mathbf{y})}_{d\lambda(\mathbf{x}, \mathbf{y}) = d(\boldsymbol{\mu} - \boldsymbol{\nu})_+ (\mathbf{x}) \, d(\boldsymbol{\mu} - \boldsymbol{\nu})_- (\mathbf{y})}$$

$$\leq 2^{p-1} \int_X d(\mathbf{x}, \mathbf{x}_0)^p \, d(\boldsymbol{\mu} - \boldsymbol{\nu})_+ (\mathbf{x})$$

$$+ 2^{p-1} \int_X d(\mathbf{y}, \mathbf{x}_0)^p \, d(\boldsymbol{\mu} - \boldsymbol{\nu})_- (\mathbf{y})$$

$$= 2^{p-1} \int_X d(\mathbf{x}_0, \mathbf{x})^p \, d|\boldsymbol{\mu} - \boldsymbol{\nu}| (\mathbf{x}). \tag{23}$$

Here, we used the fact that $a = (\boldsymbol{\mu} - \boldsymbol{\nu})_+ ([X]) = (\boldsymbol{\mu} - \boldsymbol{\nu})_- ([X])$.

**Step 6: Complete the Proof**

Combining the above estimates, we have:

$$W_p(\boldsymbol{\mu}, \boldsymbol{\nu})^p \leq 2^{p-1} \int_X d(\mathbf{x}_0, \mathbf{x})^p \, d|\boldsymbol{\mu} - \boldsymbol{\nu}|(\mathbf{x}), \tag{24}$$

which implies:

$$W_p(\boldsymbol{\mu}, \boldsymbol{\nu}) \leq 2^{\frac{1}{p'}} \left( \int_X d(\mathbf{x}_0, \mathbf{x})^p \, d|\boldsymbol{\mu} - \boldsymbol{\nu}|(\mathbf{x}) \right)^{\frac{1}{p}}. \tag{25}$$

**Lemma 7.1** When the sample space $\Omega$ is a countable set, the total variation distance between two probability distributions $\boldsymbol{\mu}$ and $\boldsymbol{\nu}$ is equal to half of their L1 norm difference [21]:

$$||\boldsymbol{\mu}, \boldsymbol{\nu}||_{TV} = \frac{1}{2} ||\boldsymbol{\mu} - \boldsymbol{\nu}||_1 = \frac{1}{2} \sum_{\omega \in \Omega} |\boldsymbol{\mu}(\omega) - \boldsymbol{\nu}(\omega)|. \tag{26}$$

**Corollary 7.1** Considering Equations (25) and (26), we have:

$$W_p(\boldsymbol{\mu}, \boldsymbol{\nu}) \leq 2^{1 - \frac{2}{p}} C_M^{\frac{1}{p}} ||\boldsymbol{\mu} - \boldsymbol{\nu}||_1^{\frac{1}{p}} = K ||\boldsymbol{\mu} - \boldsymbol{\nu}||_1^{\frac{1}{p}} \tag{27}$$

Here, $C_M$ is a constant related to the measure space. This result demonstrates that the Wasserstein distance is Hölder continuous with respect to the total variation distance.

## A.2 Statistics of datasets

**Table 4: Statistics of datasets with multi-modal item Visual (V), Acoustic (A), Textual (T) contents.**

| Dataset | Netflix | | Tiktok | | | Electronics | |
|---|---|---|---|---|---|---|---|
| Modality | V | T | V | A | T | V | T |
| Feat. Dim. | 512 | 768 | 128 | 128 | 768 | 4096 | 384 |
| User | 43,739 | | 14,343 | | | 41,691 | |
| Item | 17,239 | | 8,690 | | | 21,479 | |
| Interaction | 609,341 | | 276,637 | | | 359,165 | |
| Sparsity | 99.919% | | 99.778% | | | 99.960% | |

