# OpenReview forum: "Bridging the Gap: Teacher-Assisted Wasserstein Knowledge Distillation for Efficient Multi-Modal Recommendation"
_ACM.org/TheWebConf/2025/Conference — WWW 2025 Poster_

### Official Review · Reviewer_dGaR · 2024-11-26

**Novelty:** 4
**Technical Quality:** 5

**Review:**

## Overall

The paper introduces TARec, a framework for compressing complex multi-modal recommender systems into efficient ID-based Multi-Layer Perceptron-based Recommender systems through a teacher-assisted Wasserstein Knowledge Distillation approach, which includes a two-stage distillation process, a logit-level KD using Wasserstein Distance, and an embedding-level contrastive KD.  Extensive experiments show TARec improved efficiency and effectiveness over traditional methods.

## Pros
1. The paper is well-written and easy to understand.
2. The paper provides an anonymous code link, facilitating other researchers to reproduce and further study.

## Cons
1. The paper is missing some key information:
    1. It fails to specify which dataset was used for the Introduction section and what model yielded the presented results, so I doubt the results.
    2. The article doesn't explain the structure of the Teacher model employed, nor does it describe the structure of the teacher-assistance model or the structure of the Student model utilized.

2. In Equation 4, when calculating the logit-level distribution, since the positive sample $y_{ui}$ is used, why is it necessary to compute the negative sample part $e_u^T e_j$.

3. Experiment section lacks some key information:
    1. The results for the Teacher and TA models are not reported, making it impossible to assess the effectiveness of the methods used.
    2. The algorithm mentions the need for a well-trained MLPRec, but there is no experimental verification. It would be helpful to know how the performance would be affected without pre-training.
    3. There is a lack of comparison with some recent baselines, especially a **lightweight multi-modal recommender system**[1].

4. In my view, the main contribution of the article lies in the simultaneous use of teacher-assistance Knowledge Distillation and Wasserstein Distance-Based Knowledge Distillation, which feels more like **an assembly of two existing technologies**, lacking in innovation.
    1. The article mentions that teacher-assistance Knowledge Distillation has already been applied in some works, and the authors have not made any technical adjustments specific to the recommendation task.
    2. **The Unbiased Sinkhorn Divergence mentioned in the article is derived from [2], and the authors have failed to cite it properly and have not made any technical adjustments**. The paper [2] also contains a figure which is strikingly similar to Figure 3 in this paper.

[1] 	Xin Zhou, Zhiqi Shen. A Tale of Two Graphs: Freezing and Denoising Graph Structures for Multimodal Recommendation. ACM Multimedia 2023: 935-943

[2] S'ejourn'e, T., Feydy, J., Vialard, F., Trouv'e, A., & Peyr'e, G. (2019). Sinkhorn Divergences for Unbalanced Optimal Transport. ArXiv, abs/1910.12958.

**Questions:**

1. Please see the review above.
2. The paper employs a lightweight LightGCN model for the TA (Teacher Assistant), although the specific number of layers is not disclosed. The question arises as to why an MLP (Multi-Layer Perceptron) is used instead of continuing with the LightGCN model, and whether there is a significant difference in computational resources between the two.

**Reviewer Confidence:**

4: The reviewer is certain that the evaluation is correct and very familiar with the relevant literature

**Scope:**

3: The work is somewhat relevant to the Web and to the track, and is of narrow interest to a sub-community

---

### Official Review · Reviewer_4iFi · 2024-11-29

**Novelty:** 4
**Technical Quality:** 5

**Review:**

The author addresses two limitations of MMRecs by using a two-stage Knowledge Distillation process and replacing the original KLD with the Wasserstein Distance. I am not an expert on MMRecs, so I cannot rigorously assess the novelty of this paper. Based solely on the writing and intuition presented in this article, I believe the quality of the paper is good.

**Questions:**

There are no significant issues with this paper.

**Reviewer Confidence:**

1: The reviewer's evaluation is an educated guess

**Scope:**

4: The work is relevant to the Web and to the track, and is of broad interest to the community

---

### Official Review · Reviewer_1Ng3 · 2024-12-02

**Novelty:** 3
**Technical Quality:** 5

**Review:**

This paper introduces a two-staged distillation from cumbersome MMRecs to lightweight MLPRec, coined TARec. TARec consists of distillation from MMRec to the teacher assistant (shallower GNN) and the TA to the student MLPRec. Replacing KL divergence with the Wasserstein Distance can enable stable logit-level distillation between two distributions with large gaps. Experiments show the effectiveness and efficiency of TARec.

Pros:
1. Given the significant gap between MMRec and MLPRec, it is reasonable to propose a new distillation approach that replaces KL div with WD.
2. Extensive experiments validate the effectiveness and efficiency of TARec.
3. This paper is well-written and easy to follow.

Cons:
1. This work appears to draw upon [1], which also addresses multimodal knowledge distillation for recommendation. While the idea of replacing KL divergence with Wasserstein distance is reasonable, the novelty may seem incremental.
2. Since the TA is a shallower GNN, I wonder if it might already be efficient and effective enough for downstream tasks, potentially eliminating the need for the second distillation stage from the TA to the student. The necessity of distilling twice needs more discussion.
3. Except [1], the baselines are all before or from 2022. Some more up-to-date baselines should be compared, such as [2].
4. The figures can be further improved; removing the shadows might make it clearer.


Refs:
[1] PromptMM: Multi-modal knowledge distillation for recommendation with prompt-tuning. WWW2024

[2] LGMRec: Local and Global Graph Learning for Multimodal Recommendation. AAAI2024

**Questions:**

1. How does the first stage distillation (TA model) perform itself?
2. Can TARec surpass some other up-to-date baselines? See Con.2
3. Can you further discuss the novelty of your work regarding Con.1?
4. Since LLM distillation for RecSys has become a trending research topic, I suggest incorporating a discussion of these studies in the knowledge distilaltion for recommendation part of related works

**Reviewer Confidence:**

4: The reviewer is certain that the evaluation is correct and very familiar with the relevant literature

**Scope:**

4: The work is relevant to the Web and to the track, and is of broad interest to the community